# Towards a Better Understanding of the Relationships between Galectin-7, p53 and MMP-9 during Cancer Progression

**DOI:** 10.3390/biom11060879

**Published:** 2021-06-14

**Authors:** Yves St-Pierre

**Affiliations:** INRS-Centre Armand-Frappier Santé Biotechnologie, Laval, QC H7V 1B7, Canada; yves.st-pierre@inrs.ca

**Keywords:** galectin-7, p53, MMP-9, cancer, gain-of-function

## Abstract

It has been almost 25 years since the discovery of galectin-7. This member of the galectin family has attracted interest from many working in the cancer field given its highly restricted expression profile in epithelial cells and the fact that cancers of epithelial origin (carcinoma) are among the most frequent and deadly cancer subtypes. Initially described as a p53-induced gene and associated with apoptosis, galectin-7 is now recognized as having a protumorigenic role in many cancer types. Several studies have indeed shown that galectin-7 is associated with aggressive behavior of cancer cells and induces expression of MMP-9, a member of the matrix metalloproteinases (MMP) family known to confer invasive behavior to cancer cells. It is therefore not surprising that many studies have examined its relationships with p53 and MMP-9. However, the relationships between galectin-7 and p53 and MMP-9 are not always clear. This is largely because p53 is often mutated in cancer cells and such mutations drastically change its functions and, consequently, its association with galectin-7. In this review, we discuss the functional relationships between galectin-7, p53 and MMP-9 and reconcile some apparently contradictory observations. A better understanding of these relationships will help to develop a working hypothesis and model that will provide the basis for further research in the hope of establishing a new paradigm for tackling the role of galectin-7 in cancer.

## 1. Introduction: The Discovery of Galectin-7

The first reports on galectin-7 were published more than 25 years ago by two independent groups conducting systematic searches for differentially expressed proteins in keratinocytes. The first report was published by Prof. Julio Celis. Using two-dimensional gel electrophoresis, his group had been conducting systematic searches for differentially expressed proteins between normal and SV40-transformed human keratinocytes for several years [1]. The researchers noticed a protein, named IEF17, that was constitutively expressed in normal keratinocytes but not in transformed cells. They subsequently cloned the gene and showed that its sequence had strong homology to consensus sequences found in the carbohydrate-binding site of galectins. They further showed that the protein bound asialofetuin via a lactose-dependent interaction. Upon consultation with established galectin researchers who had just published a consensus on the criteria necessary for a protein to be recognized as a galectin the year before [2], they named this new protein galectin-7. During approximately the same period of time, a French group from L’Oréal was conducting a systematic search for epidermal-specific and stage-specific biomarkers using epidermal cDNA libraries [3]. Their attention was focused on the clone 1A12, whose sequence was identical to the sequence encoding the IEF7 protein identified by Madsen and colleagues that had just been submitted to the GenBank database. Magnaldo and colleagues constructed a GST-fusion protein with their cDNA clone, expressed it in an *E. coli* expression system and confirmed its affinity for lactose. Building on their initial discoveries, both the Danish and French groups actively pursued their research on galectin-7 and subsequently concluded that galectin-7 is a keratinocyte-specific marker found in all layers of the epidermis and other stratified epithelia of tissues, most notably in the tongue, cornea, esophagus, stomach, anus and Hassal’s corpuscles of the thymus [4,5,6,7,8]. Subsequent studies showed, however, that galectin-7 is also present in other types of epithelia, including the myoepithelial cells of the mammary gland epithelium [9,10]. In most cases, galectin-7 was found to be expressed in both the cytosolic and nuclear compartments and released into the extracellular space by a nonclassical secretion mechanism, a pattern typical of members of the galectin family [11]. Although most of the literature has focused on the expression of galectin-7 in cells of epithelial origin, it is important to keep an open mind with regards to its expression in other cell types, as we now know that it is also expressed, albeit at a rarer frequency, in cells of other lineages. We and others have reported, for example, that *LGALS7* expression is found in specific subtypes of lymphoid/myeloid cells, including transformed human B and T lymphocytes [12,13] and normal CD8-positive dendritic cells [14,15]. Too often these data are lost in the sea of transcriptomic datasets and we still largely ignore the function of galectin-7 in these normal myeloid cells.

## 2. The First Stage of the Relationship between Galectin-7 and p53

In 1997, the Bert Vogelstein group published a seminal article in Nature on p53 [16]. Similar to many teams working on p53, the Vogelstein group was studying the molecular mechanisms involved in p53-induced cell death and p53-mediated regulation of the cell cycle. They were particularly interested in identifying transcripts that were under the control of p53. For this, they used an adenoviral vector to express wild-type p53 in the p53-defective human colon cancer cell line DLD-1. They identified 14 transcripts including a transcript encoding the *p21* gene and 13 other transcripts that they called p53-induced genes (PIGs) 1–13. Many of these transcripts were functionally linked to the generation of reactive oxygen species (ROS), allowing establishment of a clear and strong connection between p53 and oxidative stress. Other transcripts encoded multiple genes, including *LGALS7* (identified as PIG1). The authors paid very little attention to galectin-7 other than a comment that this gene was part of the galectin family and that some members (such as galectin-3) were capable of inducing superoxide production in neutrophils. In fact, there were at that time no more than a handful of papers that had been published on galectin-7 (cited above). Following this article by Polyak and colleagues, several groups confirmed that upregulation of galectin-7 was associated with apoptosis in a p53-dependent manner. For example, Bernerd et al. [17] showed that galectin-7 was upregulated in skin keratinocytes upon exposure to UVB irradiation, a known inducer of the *p53* gene and apoptosis in keratinocytes. They also found that induction of galectin-7 paralleled that of wild-type (wt) p53, further confirming that induction of galectin-7, at least in UVB-exposed skin keratinocytes, is dependent on wt p53. Consistent with this hypothesis, they found no detectable expression of galectin-7 in the SCC13 keratinocyte cell line, which does not express wt p53 but rather p53 mutated at position 179 [18]. This mutation at position 179 is known to destabilize the p53 protein by altering its ability to interact with a zinc molecule, causing loss of DNA-binding specificity [19]. Our group also reported that treatment of human MCF-7 breast cancer cells (which harbor wt alleles of p53) with doxorubicin, a well-known inducer of p53, induced the expression of galectin-7 at both the mRNA and protein levels and that this induction was abrogated following knockdown of wt p53 [20]. However, after all these years, the function of galectin-7 in this physiological context related to wt p53 is still unclear. Could GAL-7 be induced by p53 to promote DNA repair? This is certainly a real possibility considering that (1) DNA repair is a pathway activated by p53, (2) galectin-7 is expressed at high levels in the nucleus, (3) galectins have been shown to have nuclear functions and (4) other members of the galectin family, such as galectin-1 and galectin-3, are known to regulate DNA damage repair [21,22,23].

## 3. A Twisted Relationship

While initial studies in colon cancer cells indicated that p53-induced galectin-7 expression was associated with cell death, a number of reports published at that time brought the universality of this phenotype into question. These studies indicated that galectin-7 might instead be associated with cancer progression (Figure 1). For example, data obtained from a study using an experimental breast carcinoma model [24] and a study evaluating clinical specimens collected from patients with thyroid cancer [25] revealed that high galectin-7 expression levels were associated with cancer progression and aggressiveness. Conducting a comparative transcriptomic analysis between a low-tumorigenic lymphoma cell line (164T2) and its aggressive variants (S11 and S19) generated by serial in vivo passaging, our group further showed that the most prominent change that occurred in highly metastatic variants was the strong upregulation (160-fold) of galectin-7 [12]. This change was unexpected, as, at that time, galectin-7 was considered a keratinocyte marker expressed in all subtypes of stratified epidermis. Analysis of gene expression profiles stored in public datasets in the Gene Expression Omnibus (GEO) repository showed that such strong upregulation of galectin-7 in lymphoid cancer cells is not unusual, as it was also observed following in vivo passaging of the parental FL5.12 cell line, a murine prolymphocytic cell line [26]. Using our lymphoma model, we further provided evidence that such upregulation of galectin-7 in lymphoma cells is not an epiphenomenon. Transfection of an expression vector containing the galectin-7 gene into low-metastatic lymphoma cells increased their metastatic behavior [27]. Conversely, inhibition of galectin-7 in aggressive lymphoma variants decreased their invasive behavior in vivo [13]. These studies established for the first time that galectin-7 can promote metastasis, a conclusion that contradicted previous observations, indicating that galectin-7 leads to cell death. However, this should have come as no surprise given previous findings showing that other members of the galectin family are well known to have a dual role in controlling cell survival depending on the cellular context, differentiation state and/or cellular localization [28,29,30]. These reports showing that galectin-7 can promote cancer progression, however, challenged the existence of a universal p53–galectin-7 axis and questioned the relevance of continuing to define galectin-7 solely as a p53-induced gene, as originally described by Polyak et al. [16]. This conclusion is supported by data showing that many cancer cells with no detectable p53 transcriptional activity, such as breast cancer cells and HaCaT cells, which carry mutations in both alleles of p53 (R282W and H179Y) that render the protein transcriptionally inactive, express high levels of galectin-7 [17,20].

## 4. De Novo Expression of Galectin-7 by Mutated p53

Although it is clear that the presence of wt p53 is not essential for a cell to express galectin-7, it is important to remember that p53 has protumorigenic functions when mutated [31,32]. Could it be that galectin-7 is induced by mutated forms of p53? Considering that p53 is mutated in approximately half of cancer cases and that galectin-7 is readily expressed in many cancer types, this is a real possibility. Our group decided to further examine this relationship between galectin-7 and p53 mutants. We studied the most frequent cancer-associated “hot spots” p53 mutations, including missense mutations within the region encoding the DNA-binding domain that involve amino acid residues in direct contact with DNA or amino acid residues that locally or globally affect the conformation of the p53 protein structure [33]. In all cases, these mutations not only eliminate the protective function normally mediated by wt p53 against stressors but also often confer cancer cells with new protumorigenic properties through a gain-of-function (GOF) mechanism. The GOF mechanism was first proposed by Arnold Levine [34]. Using a series of vectors encoding either mouse or human mutant p53 transfected into p53^null^ cells, new or additional phenotypes were conferred on these cells, suggesting that cancer cells with mutant p53 are likely to be more aggressive and have a poorer prognosis than cancer cells with no p53. This suggested that mutant p53 not only acts through a dominant-negative effect on the remaining wt p53 allele but can also trigger the selection of clonal cells with new properties. This could occur through at least two nonexclusive transcriptional and non-transcriptional mechanisms, thereby altering specific signaling pathways [35]. This is particularly well documented in the case of the NF-κB pathway, which is activated by mutant p53, causing chronic inflammatory conditions that favor the onset of transformation [36]. Using expression vectors encoding specific p53 mutants, we found that R175H induces de novo expression of galectin-7 in p53^null^ human breast (MDA-MB-453) and ovarian (SKOV-3) cancer cell lines [20,37]. This does not, however, mean that p53 mutants universally induce galectin-7 in every cancer cell line. The cellular context is likely to play a role. For example, another mutant, R273H, induces galectin-7 in breast cancer cells but not in ovarian cancer cells [20,37]. We also found that treatment of MDA-MB-231 (R280K) and MCF-7 (harboring wt p53) breast cancer cells with doxorubicin, a widely used chemotherapeutic agent, induced galectin-7 in both cell lines. In fact, we found that both can bind to the *LGALS7* promoter in breast cancer cells [20]. It is thus logical to suggest that both wild-type and mutant p53 can induce galectin-7 in breast cancer cells, but given the intrinsic resistance of these cells to cell death, the balance is shifted towards the protumorigenic role of galectin-7 (Figure 2). Interestingly, we found that R273H-induced expression of galectin-7 was inhibited by caffeic acid phenethyl ester (CAPE) and quercetin, two inhibitors of NF-κB [20,38,39]. It is important to note, however, that constitutive NF-κB activity is not sufficient to induce galectin-7. For example, p53-null MDA-MB-453 cells exhibit high levels of NF-κB activity but do not express detectable levels of galectin-7.

Given the multiple roles of galectins in controlling the tumor microenvironment, it is reasonable to hypothesize that the production and release of galectin-7 by cancer cells is one of multiple mechanisms used by GOF mutants of p53 to facilitate a protumorigenic microenvironment [40]. Such a role for galectin-7 in the tumor microenvironment would include its ability to induce local immunosuppression, as we know now that galectin-7, like many galectins, induces apoptosis in activated immune cells [37,41,42]. This role would also include the regulation of the expression of protumorigenic genes in cancer cells. In this context, a number of studies have established a new functional link between galectin-7 and MMP-9, a member of the zinc metalloproteinases family involved in the degradation of the extracellular matrix and several steps of the metastatic process [43]. Such an association between galectin-7 and MMP-9 has been observed in many cancer types. It was first reported in an experimental lymphoma mouse model, in which the expression of galectin-7 by lymphoma cells was shown to increase the aggressive behavior of lymphoma cells by inducing MMP-9 [27]. We and others have since shown that galectin-7 can induce MMP-9 in breast cancer, ovarian cancer, head and neck cancer, cervical and oral squamous cell carcinomas and, more recently, gastric cancer cells [44,45,46,47,48,49]. In some cases, this association between galectin-7 and MMP-9 was found to be specific, as no such association was found between MMP-9 and galectin-1 or galectin-3 [44,45]. Today, when studying the role of galectin-7 in tumor progression, most notably during metastasis, it is difficult to ignore the relationship of galectin-7 with MMP-9.

## 5. How Does Galectin-7 Induce MMP-9?

Increased expression of MMP-9 induced by galectin-7, which occurs at both the mRNA and protein levels, has in many cases been documented by adding recombinant human galectin-7 to cancer cells. The use of this approach is largely due to the fact that galectin-7 is relatively easy to produce and purify by standard lactosyl-Sepharose affinity chromatography [41,50]. We can propose a model that integrates the multiple findings on the mechanisms by which galectin-7 can induce MMP-9 (Figure 3). Binding of recombinant galectin-7 to specific cell surface receptors can trigger a number of signaling pathways, including the p38 MAPK, ERK1/2 and JNK pathways [41,47]. The use of pharmacological inhibitors, such as SB203580, an inhibitor of the p38 pathway, and inhibitors of the ERK1/2 and JNK pathways, has shown that both pathways are likely important for galectin-7-mediated induction of MMP-9 [44,47]. This is not very surprising, as these pathways are commonly used by growth factors to induce MMP-9 expression in cancer cells [43]. The ability of galectin-7 to augment MMP-9 expression in cancer cells has also been established using cancer cells transfected with eukaryotic expression vectors containing the gene encoding galectin-7 (i.e., *LGALS7*). Given the propensity of galectins to be released into the extracellular microenvironment, the ability of galectin-7 to induce MMP-9 expression is again likely to occur via extracellular binding of galectin-7 to cell-surface receptors. This hypothesis is supported by data showing that the increased MMP-9 expression induced by galectin-7 transfectants is inhibited by the addition of lactose to the culture medium [27]. However, another possible pathway that may contribute to the increased expression of MMP-9 induced by galectin-7 involves galectin-7 re-entry by endocytosis via clathrin-coated pits in a manner similar to that reported for other galectins [51]. This intracellular pool of galectin-7 is targeted by a relatively large diversity of binding partners depending on the cell type [41]. This has been well documented in the case of galectin-7/Bcl-2 interactions [52]. Galectin-7 is also targeted by Tidf1, which represses the oncogenic role of galectin-7 by promoting its degradation [49]. Whether de novo expression of Tidf1 represses galectin-7-induced MMP-9 expression is an interesting possibility that has not, to our knowledge, been studied.

The main question that remains is what cell-surface glycoreceptors bind galectin-7 and trigger intracellular signaling that ultimately leads to MMP-9 expression? Given the promiscuity of galectins for multiple glycosylated cell-surface receptors, it is likely that the signals implicated in the upregulation of MMP-9 involve multiple receptors that vary according to the cancer subtype. Accordingly, a logical hypothesis is that extracellular galectin-7 induces MMP-9 via stabilization and/or reorganization of cell-surface glycoreceptors to lower the signaling threshold necessary for triggering intracellular signaling [53]. This has been well described for galectin-1 and galectin-3 [54,55,56]. We must also be open to the possibility that such a signaling cascade could in fact be initiated by carbohydrate-independent interactions. This has been well documented in the case of galectin-1, which binds to unglycosylated membrane pre-B cell receptor, which regulates glycan-dependent formation of lattice [57]. A recent study indeed showed that galectin-7 could bind to nonglycosylated cell-surface receptors such as E-cadherin [58]. The binding of galectin-7 to E-cadherin, which normally undergoes endocytosis, stabilizes the receptor at the membrane and restrains its endocytosis. Interestingly, E-cadherin is a well-known substrate of MMPs, including MMP-9 [59,60,61,62,63]. E-cadherin being one of these critical receptors that induces MMP-9 in cancer cells is a real possibility. In such a case, the cleavage of E-cadherin by MMP-9 would ensure strict autoregulation of the galectin-7–MMP-9 axis. Future investigations will be necessary to test this hypothesis.

## 6. Conclusions

The objective of this review was to shed light on the relationships between galectin-7, p53 and MMP-9. It is clear that we should avoid referring to galectin-7 simply as a p53-induced gene for two reasons. First, galectin-7 is not always associated with apoptosis and is often, if not most of the time, associated with tumor progression. Second, there is no direct relationship between galectin-7 expression and p53 transcriptional activity. Many p53^null^ cells express high levels of galectin-7, while transcriptionally inactive mutant p53 induces galectin-7. However, it may be too early to question the relationship between wt p53 and galectin-7. Indeed, it is important to note that wt p53 can behave like a mutant form if its structural integrity is not properly maintained by a chaperone that prevents misfolding into a mutant-like form [64]. The reverse is also true, as mutant p53 can be reverted into a conformationally wt-like form of p53 [65]. In contrast, galectin-7 seems to have a strong and long-lasting relationship with MMP-9. Clearly, both can be induced by similar mechanisms, including the GOF activity of mutant p53. This relationship among galectin-7, MMP-9 and mutant p53 could be more common than we think, considering the increasing number of studies showing that galectin-7 promotes cancer progression. Future studies should also pay attention to another member of the p53 family, i.e., p63, which has an expression profile close to that of galectin-7 (stratified and glandular epithelia) [66]. Moreover, it has been reported that galectin-7 could regulate keratinocyte proliferation and differentiation through c-Jun N-terminal kinase (JNK1)-miR-203-p63 pathway [67].

## Figures and Tables

**Figure 1 biomolecules-11-00879-f001:**
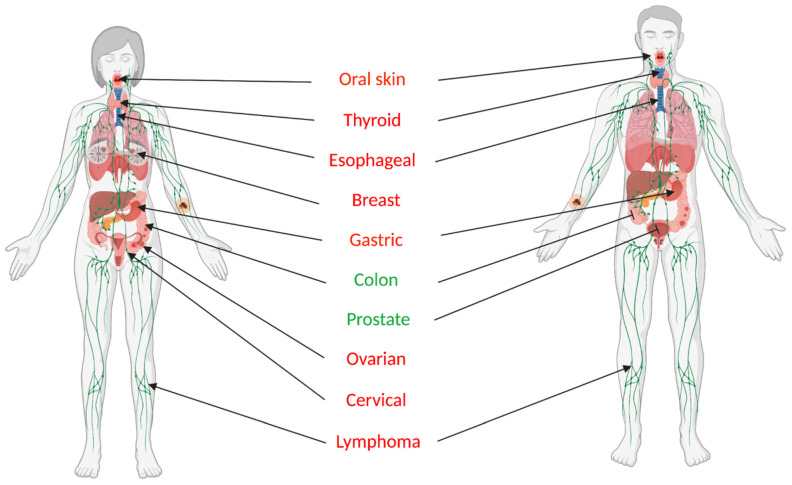
The dual role of galectin-7 in cancer. In most cancer types, overexpression of galectin-7 is associated with and/or promotes cancer progression. For some types of cancer, such as colon cancer and prostate cancer (in green), expression of galectin-7 is rather associated with an anti-tumor role.

**Figure 2 biomolecules-11-00879-f002:**
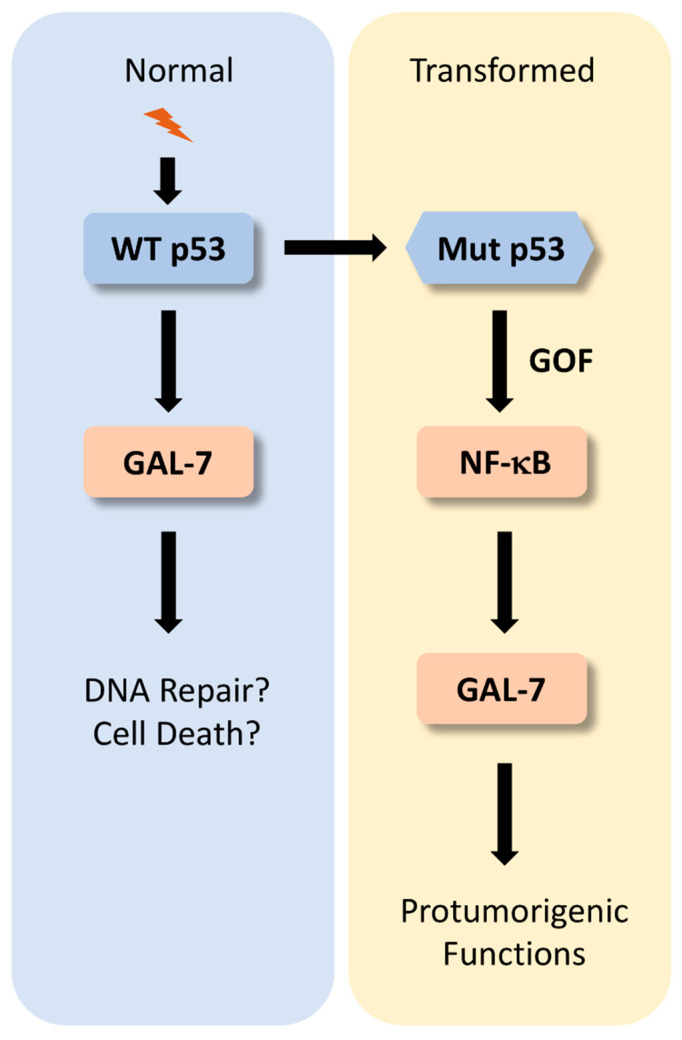
Reconciling the relationship between galectin-7 with wt and mutant p53. In normal cells harboring wt p53 alleles, galectin-7 is induced by de novo expression induced by wt p53 following stress signals. This pathway is also active in some cancer types, such as colon cancer [16]. Expression of nuclear galectin-7 may then regulate the cell cycle and/or DNA repair. In cancer cells with intrinsic resistance to cell death, galectin-7 can be induced by a gain-of-function (GOF) mechanism via mutant p53, shifting the balance towards the pro-tumorigenic role of galectin-7.5. The galectin-7–MMP-9 relationship in cancer.

**Figure 3 biomolecules-11-00879-f003:**
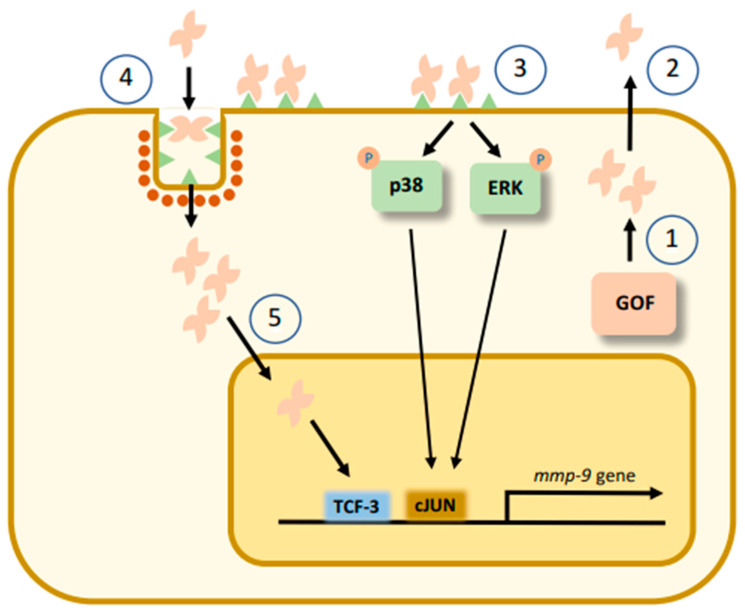
A working model for the control of MMP-9 gene expression in cancer cells via galectin-7. De novo expression of galectin-7 in cancer cells by the GOF pathway leads to accumulation of intracellular galectin-7 (1). Intracellular galectin-7 translocates directly to the nucleus to trigger MMP-9 gene expression (5) or be released by cancer cells in the extracellular tumor microenvironment (2). Once outside cancer cells, extracellular galectin-7 can bind and activate glycoreceptor-mediated signaling pathways that lead to MMP-9 gene expression (3). Such a mechanism can be activated in adjacent cancer cells or in the same cell following an autocrine mechanism (4).

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
