# Peer review of "Towards a Better Understanding of the Relationships between Galectin-7, p53 and MMP-9 during Cancer Progression"

_biomolecules, 2021, doi:10.3390/biom11060879_

Round 1

Reviewer 1 Report

Overall, this is a very interesting and well-written paper. The author organizes the review by taking the reader through a historical overview through to an outline of interesting work that still needs to be done, and I found this helpful and thought-provoking. It is certainly appropriate for publication in Biomolecules.

The author should add a section providing information on the structure of galectin-7 and any known molecular-level information about interactions between MMPs and galectin-7 as well as between galectin-7 and p53. For example, an MMP cleavage site on the N-terminal domain has been proposed; is there anything comparable on galectin-7? Binding of some galectins to p53 has been reported to enable phosphorylation at specific amino acids on p53; is there anything comparable known about galectin-7? If nothing has yet been studied at the molecular level for galectin-7 interactions with MMP-9 or p53, then what does the author hypothesize based on comparison and contrast of the structure of galectin-7 to other galectins? At this point, the complete lack of information on the molecular level up to quaternary structure of galectin-7 and what this leads us to expect about the relationship between galectin-7, p53, and MMP is not included in the article.

Author Response

Reply: The overall objective of the review is focused on the functional relationships between all three molecules. Although it is true that other members of the galectin family can act as substrates for MMP’s, this is not the case for GAL-7, which does not harbor a polypeptide link between both CRDs, as in the case of galectins of the tandem repeat subclass. Moreover, other reviews have focused on the overall quaternary structure of GAL-7 and other galectins (Kamitori et al., 2018; St-Pierre et al., 2018).

Reviewer 2 Report

The manuscript is interesting and relevant for the field. However, there are some minor issues that need to be addressed before in order to enrich the manuscript before publication:

1) In section "3. A twisted relationship", the author should include a table or scheme about the known contexts/diseases where Galactin 7 expression is good or bad. This would be very useful for the reader;

2) Figure 1 and 2 should be cited in the text;

3) Figure 2 should be on section 6 instead of section 4;

Author Response

Reply: I would like to thank the reviewer for his/her suggestion. A figure has been added in the revised version of the manuscript regarding the dual role of galectin-7 in cancer. All figures are now cited in the text.

Reviewer 3 Report

In the present review, the author investigated the relationship between galectin-7, p53 and MMP9. He first established an inventory of knowledge on galectin-7, including its discovery and its classification as a p53-induced gene. Then, he listed numerous examples of galectin-7 overexpression in the absence of wtp53, in particular in cancer cells harboring a mutated p53. A discussion follows on the dual effects associated with galectin-7 expression depending of the cellular context, i.e. cell death or cancer progression and aggressiveness.

The second part of the review was dedicated to the relationship between galectin-7 and MMP9 expression, as seen in several cancer types. The molecular mechanisms, as well as the biological consequences on cancer progression, were listed and discussed.

This is a well-written review that provides information and clarification on the complex relationship between p53-galectin-7 and MMP9. The presentation, which takes up the history of the discovery, is interesting and original. Nevertheless, an important partner has been forgotten, the p63 protein. Indeed, this p53 family member has an expression profile close to those of GAL-7 (stratified and glandular epithelia). Moreover, it has been reported that galectin-7 could regulate keratinocyte proliferation and differentiation through a JNK1-miR-203-p63 pathway (DOI: 10.1038/JID.2015.366). The potential implication of p63 in the expression and role of galectin-7 should be addressed.

Author Response

Reply: I would like to thank the reviewer for his/her suggestion to add the potential link between galectin-7 and p63. The comment, along with two references (#66 and 67), have been added in the revised version of the manuscript.